# Towards Stricter Black-box Integrity Verification of Deep Neural Network Models

## ABSTRACT

Cloud-based machine learning services are attractive but expose a cloud-deployed DNN model to the risk of tampering. Black-box integrity verification (BIV) enables the owner or end-users to ascertain whether a cloud-deployed DNN model has been tampered with via returned responses of only top-1 labels. Fingerprinting generates fingerprint samples to query the model to achieve BIV of the model with no impact on the model's accuracy. In this paper, we introduce BIVBench, the first benchmark for BIV of DNN models, encompassing 16 types of practical modifications covering typical tampering scenarios. We reveal that existing fingerprinting methods, which focus on a limited range of tampering types, lack sensitivity in detecting subtle, yet common and potentially severe, tampering effectively. To fill this gap, we propose MiSentry (Model integrity Sentry), a novel fingerprinting method that strategically incorporates only a few crucial subtly tampered models into a model zoo, leverages meta-learning, and maximizes the divergence of the output predictions between the untampered targeted model and those models in the model zoo to generate highly sensitive, generalizable, and effective fingerprint samples. Extensive evaluations using BIVBench demonstrate that MiSentry substantially outperforms existing state-of-the-art fingerprinting methods, particularly in detecting subtle tampering.

## KEYWORDS

integrity verification, deep neural networks, meta-learning

## 1 INTRODUCTION

Deep neural networks (DNNs) have significantly advanced the state of the art for many computer vision tasks [18, 37, 47]. To facilitate wide use of DNN models, major cloud operators provide machine-learning platforms, such as Microsoft Azure ML [40], Google AutoML [15], and Amazon SageMaker [2], to allow customers to deploy their DNN models in the cloud. However, deploying DNN models in the cloud brings several risks. An adversary may tamper with a deployed DNN model via Trojan [36] or backdoor [17] attacks to insert harmful behavior. An unethical model provider might sabotage a competitor's model, such as through a bit-flipping [61] attack, to degrade its performance to gain a competitive advantage. A dishonest cloud service provider may stealthily replace a model with a simpler one, such as a compressed model, to reduce operational costs.

To mitigate these threats, both *fingerprinting* [22, 58] and *watermarking* [64] are proposed for *black-box integrity verification (BIV)* of DNN classification models. BIV aims to detect model tampering with only black-box access to the deployed model, i.e., by querying

a deployed model which returns only top-1 category labels as responses. The model is detected as being modified if any response to the querying samples disagrees with its ground truth, which is the label returned by *the target model* (i.e., the unmodified model to protect).

Watermarking embeds watermark samples into the model through training or fine-tuning before deployment. Thus it can potentially reduce the model's performance. Moreover, the finite number of pre-embedded watermark samples exposes its vulnerability to watermarking removal where adversaries could remove watermarking by exploiting leaked samples.

In contrast, fingerprinting does not alter the model and enables the unlimited generation of fingerprint samples at any time, allowing for its resilience to leakage. Therefore, fingerprinting offers a more robust and flexible solution for black-box tampering detection of DNN models compared to watermarking, making it generally the superior choice for ensuring the integrity of deployed models.

However, existing fingerprinting methods are evaluated on limited types of model tampering, including model pruning, Trojan, and backdoor attacks, which are all *obvious malicious modifications*. But model tampering could occur in countless ways, subtle or obvious, benign or malicious. *Subtle model modifications* are also common. For example, an adversary can launch subpopulation data-poisoning attacks [27] to compromise the performance of a DNN model on a particular small subset of samples of interest. Furthermore, a DNN model can also be evolving or adapting over time in benign ways [14, 50], which are *benign modifications*. When dealing with multiple versions of a model that has been updated or tuned for different purposes, the service provider may unintentionally operate with a mismatched version of the DNN model. Thus it is practically necessary to verify if a cloud-serviced model is the correct version.

In traditional integrity verification of emails and files, it assures that data is unaltered from its original form. Similarly, integrity verification of a DNN model should detect any modifications to the model, malicious or benign, significant or subtle. However, existing works only focus on detecting malicious significant modifications and ignore others. Stricter integrity verification of DNN models is left to be explored. We take the first step towards such an ultimate goal with the following research question: black-box integrity verification of DNN models for *harder-to-observe model modifications*.

A model is said to be *observably modified* if its decision boundary is observably altered, i.e., there exists an example such that its prediction result received by an end user, e.g., top-1 label for a classification model, is observably different from that of the target model. Intuitively, subtler model modifications are harder to observe. Stricter integrity verification for detecting harder-to-observe modifications is useful, since an end user may need to check if the model he/she is using is really the model he/she wants to use.

To fill this gap, we introduce *BIVBench*, the first benchmark for black-box integrity verification of DNN models. *BIVBench* comprises 16 types of *practically observable* model modifications, capturing common tampering scenarios and including 10 new tampering types previously unexplored. Our empirical evaluations conducted using BIVBench reveal that existing SOTA fingerprinting methods are less effective, or even ineffective, in detecting harder-to-observe subtle tampering (see Section 3.2 for details). Detecting harder-to-observe model modifications such as subtle model modifications requires fingerprint samples with higher sensitivity.

We then propose *MiSentry* (Model integrity Sentry), an innovative DNN fingerprinting method with higher sensitivity empowering stricter black-box integrity verification. By maximizing the divergence of the output predictions between the target model and potential tampered models (called model zoo), MiSentry generates fingerprint samples capable of effectively distinguishing seen model tampers. Inspired by our novel empirical observations that fingerprint samples generated with subtler modifications can detect more obvious modifications and exhibit higher detection sensitivity, MiSentry meticulously selects representative models of only minor modification types hard to distinguish from the target DNN model for constructing the model zoo to maximize the sensitivity of generated fingerprint samples. In addition, MiSentry leverages meta-learning [62] with the model zoo to generate fingerprint samples that can generalize well to unseen tampering types. Extensive experiments on BIVBench validate the high sensitivity and generalizability of fingerprint samples generated by our MiSentry. It outperforms existing fingerprinting methods for DNN models, esp. for subtler malicious tampering and benign modifications.

Our major contributions can be summarized as follows:

- We are the first to explore black-box integrity verification of DNN models in a stricter form by defining observable modification and considering subtle tampering.
- We introduce *BIVBench*, the first benchmark specifically designed for black-box integrity verification of DNN models. This benchmark encompasses 16 types of *practically observable* modifications capturing all prevalent scenarios of potential DNN model tampering. This is a notable broadening in scope compared to existing works, which focus on no more than 6 types of model tampering. Our empirical analyses with BIVBench reveal that existing SOTA fingerprinting methods are insufficiently sensitive in detecting such subtle tampering.
- To effectively distinguish all practically observable model modifications, we propose *MiSentry*, a novel fingerprinting method that strategically incorporates only a few crucial subtly tampered models into a model zoo, leverages meta-learning, and maximizes the divergence of the output predictions between the untampered targeted model and models in the model zoo to generate highly sensitive, generalizable, and effective fingerprint samples.
- Extensive experimental evaluation using BIVBench confirms that MiSentry can effectively detect observable modifications, including those unseen during sample generation. MiSentry consistently outperforms all existing methods, particularly in detecting subtle tampering where existing SOTA fingerprint methods are less effective or even ineffective.

## 2 BLACK-BOX INTEGRITY VERIFICATION

Both watermarking [64] and fingerprinting [22] are proposed for black-box integrity verification of a DNN model through querying it with responses of only top-1 category labels.

Sensitive-Sample Fingerprinting (SSF) [22] is the first fingerprinting method to detect tampering with DNN models. It generates sensitive and natural-looking samples by approximating and maximizing their sensitivity to the model's tampering, and selects a subset of sensitive samples with the maximal coverage rate of activated neurons as fingerprint samples. Symbolic constraint solvers are used in [12] and Bayesian optimization (BO) is used in [34] to solve SSF's optimization problem in generating sensitive samples. PublicCheck [58] generates fingerprint samples located near the decision boundary with the generative models in a black-box manner, since a model's decision boundary can be sensitive to tampering.

All the existing fingerprinting methods focus on detecting backdoor attacks and model compression but ignore subtler model tampering. Model tampering could happen more covertly and subtly. We refer to model tampering with subtler modifications to the decision boundary of the target model than the model tampering types discussed in existing literature, such as SSF and PublicCheck, as *subtle model tampering*. In this paper, we incorporate several types of subtle modifications that may occur in practice into BIVBench. These include malicious ones, such as sample-level attacks (Targeted Attack, Degradation-S), and benign ones, like online learning, unlearning, and fine-tuning of the last layer or all layers (see Sec. 3.1 and Appendix C.2 for more details). As to be presented in Section 5, existing fingerprinting methods lack sensitivity or are ineffective in detecting such subtler model tampering, while our MiSentry demonstrates its effectiveness in detecting such subtle tampering.

Furthermore, unlike SSF-like methods which estimate the fingerprint's sensitivity to tampering through a series of relaxations, our MiSentry directly assesses sensitivity with prediction changes following potential subtle tampering, without any relaxation, thereby allowing for higher sensitivity. Also, in contrast to PublicCheck-like methods where fingerprint samples are located near the decision boundary and could exhibit unusual prediction vectors which can be easily figured out when querying, our MiSentry maximizes output covertness by ensuring prediction vectors of generated fingerprint sample similar to those of normal samples while simultaneously maximizing sensitivity.

Model hashing [4, 28, 48] and watermarking techniques [3] also aim for integrity verification, but they operate differently and cater to distinct threat models from ours. Specifically, model hashing requires white-box access to the protected model, which contrasts with our methodology which assumes black-box query access during integrity verification. The black-box approach is more challenging but practical in real-world scenarios. In addition, model watermarking involves tampering with the target model, which could result in performance degradation. MiSentry, however, does not tamper with the target model.

The primary advantage of black-box integrity verification over white-box integrity verification is its ability to conduct live and stealthy integrity verification while the model is in service, without being detected by the cloud service provider. This makes it difficult for a dishonest service provider to evade black-box integrity

verification. In contrast, white-box integrity verification requires white-box access to the model for integrity verification, rather than during the model's service. This allows a dishonest service provider to provide the authentic model for white-box integrity verification while using a compressed model for servicing users. Black-box integrity verification makes integrity verification more feasible, stealthy, practical, and harder to evade.

Black-box integrity verification is more challenging to design than white-box integrity verification since only the top-1 label is available for integrity verification, making the detection of subtle changes significantly more difficult.

Fingerprinting is also used for intellectual property (IP) protection of DNN models [5, 44, 59, 60]. IP protection aims to verify the ownership of a stolen DNN model. Fingerprint samples for IP protection should be robust to model modifications that can be employed to obfuscate the model's ownership. In contrast, integrity verification ensures that a model operates as intended and has not been altered. IP protection and integrity verification pursue distinct objectives and possess different characteristics.

## 3 BIVBENCH: BENCHMARK FOR BLACK-BOX INTEGRITY VERIFICATION OF DNNS

Existing works only evaluate the performance of black-box integrity verification on limited types of model tampering, including model pruning, Trojan, and backdoor attacks. However, model tampering could occur in countless ways. Subtle and benign model modifications are common and their detection is also crucial. To cover common tampering scenarios and fill the gap of detection for unexplored tampering types, we introduce *BIVBench*, the first benchmark for black-box integrity verification of DNN models.

### 3.1 Practical Model Modifications to DNNs

Traditional integrity verification of emails and files requires that any modifications to the data, malicious or benign, should be detected. Far from such much stricter integrity verification, existing fingerprinting methods mainly consider limited types of obvious malicious model which are relatively easier to detect. We aim to enable integrity verification of a DNN model to detect *any observable model tampering*, where the decision boundary of the model is altered such that there exists an example whose prediction is observably different from that of the original model. Such integrity verification should be able to detect any practical modifications, benign or malicious, to a DNN model. We come up with the following practical modifications that a fingerprinting method should be able to detect.

**Benign Modifications**. DNN models may be modified for legitimate reasons. These benign modifications include:

- **Unlearning** [14] is to force a model to forget a selected subset of data used to train the model. This is increasingly important with the enactment of privacy protection laws such as GDPR.
- **Online learning** [50] is used to adapt a model to fit newly-collected incremental data.
- **Model compression** [7] compresses a DNN model to save computing resources, typically in the following ways:
  - (i) **Pruning** [41] prunes components with the least contribution to the performance of the model and then retains the lost performance by fine-tuning the rest.

  - (ii) **Quantization** quantizes a model, i.e., reducing the precision of the model's representation.
- **Knowledge distillation** [23] transfers the knowledge from a teacher model to a student model. By transferring to a smaller model, it also reduces computing resources.
- **Transfer learning** [43] adapts a pre-trained model to a new downstream task.
- **Fine-tuning** adjust the weights of specific layers (called *tunable layers*) while freezing the weights of other layers.
- **Retraining** differs from fine-tuning by re-initializing the weights of tunable layers before relearning them.

**Malicious Modifications**. A model can be maliciously modified via poisoning training data or directly modifying the model to degrade its performance or misbehave only on specific inputs. Malicious modifications include:

- **Poisoning degradation attack** [27] degrades the accuracy of a model by tampering with its training data.
- **Bit-flipping attack** [46] degrades the accuracy of a model by flipping the most vulnerable bits of the model.
- **Targeted attack** [30, 51, 54] makes a model mispredict to a specific label on specific input samples but predict normally on other input samples.
- **Backdoor attack** [16, 29] makes a model mispredict to a specific label when a specific trigger is applied to an arbitrary sample but behaves normally otherwise. A backdoor can be inserted into a model by poisoning its training data.
- **Trojan attack** [36] injects a backdoor into a model by first optimizing the values of a backdoor trigger at a given location without touching the model and then applying the trigger to fine-tune ending layers.
- **Clean-label attack** [63] inserts a backdoor using poisoning data that visually appears consistent with the clean labels.

We construct BIVBench, the first benchmark for DNN black-box integrity verification, with all the benign and malicious modifications listed in the above practical model modifications.

### 3.2 Fingerprinting Evaluation Using BIVBench

We next comprehensively evaluate black-box integrity verification performance of existing SOTA fingerprinting methods using our BIVBench. The performance is metricized by the tampering detection rate using $N_S$ fingerprint samples for each tampering type. Three frequently-used datasets, CIFAR-10 [31], GTSRB [53], and ImageNet [49], and models with various architectures are used. Specifically, we use ResNet20 [19], CNN [35] with 6 convolution layers and 1 full-connected layer, and DenseNet121 [26] as the architectures of the target model for CIFAR-10, GTSRB and ImageNet, respectively. Test instances for each tampering type are constructed following the principle of *subtlest tampering* that tampering action is stopped as soon as the tampering objective is met since subtler modifications are harder to detect. We use the existing state-of-the-art (SOTA) fingerprinting methods, SSF [22] and PublicCheck [58], as the baseline methods. Implementation details about fingerprint generation, tampering test instances, models, and tampering detection are described in Appendix C.

The tamper detection rates of SSF, PublicCheck, and MiSentry (which we will propose in the next section) for different tampering types using 1 ($N_S = 1$) and 5 ($N_S = 5$) fingerprint samples for

Table 1: Tamper detection rate (%). $N_S$ is the number of fingerprint samples used for integrity verification. N/A means not available. For fine-tuning and retraining, the subtext indicates if the last layer (Last) or all layers (All) are fine-tuned, and the number after dash (e.g., 3) means the negative power of the learning rate (e.g., $10^{-3}$) used in fine-tuning. For degradation (poisoning degradation attack), the letter after the dash indicates if a single sample (S) or all the samples in a randomly selected category (C) are mislabeled. For the latter, the subtext indicates if these samples are randomly mislabeled (Random) or mislabeled to a specific label selected (Specific).

| Dataset | CIFAR-10 | | | | | | GTSRB | | | | | | ImageNet | | | | | |
|---|---|---|---|---|---|---|---|---|---|---|---|---|---|---|---|---|---|---|
| Method | SSF | | PubCheck | | MiSentry | | SSF | | PubCheck | | MiSentry | | SSF | | PubCheck | | MiSentry | |
| Tampering Type \ $N_S$ | 1 | 5 | 1 | 5 | 1 | 5 | 1 | 5 | 1 | 5 | 1 | 5 | 1 | 5 | 1 | 5 | 1 | 5 |
| Unlearning | 3.2 | 20.9 | 28.3 | 86.6 | **68.4** | **99.8** | 1.9 | 16.4 | 19.9 | 73.5 | **59.4** | **97.8** | 39.6 | 94.1 | 56.3 | 97.8 | **87.7** | **100.0** |
| Online Learning | 6.7 | 36.8 | 25.4 | 82.3 | **74.1** | **100.0** | 4.2 | 35.7 | 21.8 | 69.7 | **63.7** | **99.5** | 46.1 | 96.3 | 48.2 | 96.9 | **89.8** | **100.0** |
| Pruning | 66.3 | 99.7 | 69.5 | 100.0 | **89.5** | **100.0** | 57.1 | 97.5 | 74.6 | 100.0 | **84.5** | **100.0** | 64.8 | 98.3 | 74.3 | 100.0 | **92.4** | **100.0** |
| Quantization | 66.7 | 99.8 | 60.8 | 99.4 | **90.7** | **100.0** | 71.5 | 100.0 | 71.2 | 100.0 | **91.2** | **100.0** | 84.5 | 100.0 | 81.9 | 100.0 | **95.3** | **100.0** |
| Knowledge Distillation | 78.4 | 100.0 | 69.2 | 100.0 | **92.8** | **100.0** | 76.2 | 99.9 | 68.3 | 100.0 | **88.4** | **100.0** | 82.7 | 100.0 | 88.7 | 100.0 | **94.9** | **100.0** |
| Transfer Learning | 85.9 | 100.0 | 74.3 | 100.0 | **92.3** | **100.0** | N/A | N/A | N/A | N/A | N/A | N/A | N/A | N/A | N/A | N/A | N/A | N/A |
| Fine-tuning$_{Last}$-3 | 4.6 | 36.1 | 27.9 | 89.4 | **73.7** | **100.0** | 2.2 | 18.3 | 18.6 | 71.4 | **61.1** | **99.2** | 44.9 | 95.2 | 47.1 | 96.3 | **88.1** | **100.0** |
| Fine-tuning$_{Last}$-4 | 1.8 | 15.3 | 24.8 | 83.2 | **58.1** | **98.8** | 0.5 | 9.1 | 16.8 | 66.9 | **42.5** | **97.1** | 35.5 | 92.7 | 41.3 | 91.8 | **83.4** | **100.0** |
| Fine-tuning$_{Last}$-5 | 0.0 | 0.0 | 21.6 | 79.9 | **45.9** | **96.2** | 0.0 | 0.0 | 16.1 | 67.2 | **36.8** | **90.3** | 31.1 | 89.5 | 39.9 | 92.4 | **82.3** | **99.8** |
| Fine-tuning$_{All}$-3 | 7.3 | 38.2 | 29.5 | 88.5 | **76.5** | **100.0** | 5.4 | 37.5 | 21.2 | 78.5 | **64.9** | **99.5** | 47.2 | 97.1 | 54.4 | 98.9 | **90.7** | **100.0** |
| Fine-tuning$_{All}$-4 | 2.4 | 19.6 | 27.1 | 86.8 | **64.1** | **99.4** | 1.7 | 14.6 | 17.4 | 69.3 | **58.2** | **96.4** | 37.4 | 93.5 | 45.8 | 96.1 | **87.5** | **100.0** |
| Fine-tuning$_{All}$-5 | 0.0 | 0.0 | 26.2 | 85.3 | **59.2** | **97.6** | 0.0 | 0.0 | 17.2 | 65.8 | **44.5** | **94.8** | 35.8 | 90.6 | 40.5 | 92.7 | **83.2** | **100.0** |
| Retraining$_{Last}$ | 66.5 | 99.7 | 39.1 | 91.7 | **84.5** | **100.0** | 59.3 | 95.8 | 58.5 | 98.4 | **78.1** | **99.9** | 67.4 | 99.8 | 78.1 | 100.0 | **93.6** | **100.0** |
| Retraining$_{All}$ | 76.1 | 100.0 | 65.4 | 99.3 | **92.6** | **100.0** | 64.5 | 97.3 | 64.8 | 99.1 | **79.6** | **100.0** | 68.0 | 100.0 | 82.0 | 100.0 | **94.0** | **100.0** |
| Different Architectures | 79.7 | 100.0 | 65.1 | 99.2 | **93.5** | **100.0** | 67.4 | 99.2 | 68.7 | 100.0 | **81.9** | **100.0** | 78.3 | 100.0 | 84.5 | 100.0 | **95.1** | **100.0** |
| Degradation$_{Random}$-C | 43.9 | 92.5 | 48.4 | 94.7 | **91.3** | **100.0** | 49.6 | 96.5 | 53.5 | 95.8 | **85.3** | **100.0** | 61.3 | 97.4 | 73.4 | 99.9 | **91.5** | **100.0** |
| Degradation$_{Specific}$-C | 46.2 | 95.2 | 47.9 | 94.5 | **93.1** | **100.0** | 55.3 | 97.4 | 59.2 | 97.9 | **92.7** | **100.0** | 64.5 | 98.9 | 72.9 | 100.0 | **91.6** | **100.0** |
| Degradation-S | 19.8 | 72.4 | 38.7 | 91.1 | **88.2** | **100.0** | 26.1 | 85.7 | 38.9 | 92.2 | **83.4** | **100.0** | 56.9 | 96.8 | 67.3 | 99.8 | **92.3** | **100.0** |
| Targeted Attack | 11.4 | 63.5 | 36.9 | 90.2 | **83.8** | **100.0** | 12.4 | 67.1 | 42.1 | 94.4 | **79.4** | **100.0** | 48.1 | 94.9 | 63.8 | 99.8 | **90.9** | **100.0** |
| Bit-flipping | 84.0 | 100.0 | 91.0 | 100.0 | **91.0** | **100.0** | 89.0 | 100.0 | 88.0 | 100.0 | **92.0** | **100.0** | 96.0 | 100.0 | 96.0 | 100.0 | **98.0** | **100.0** |
| Backdoor | 64.9 | 98.8 | 73.6 | 99.6 | **86.4** | **100.0** | 63.2 | 98.6 | 69.3 | 99.7 | **85.8** | **100.0** | 65.6 | 99.8 | 81.4 | 100.0 | **92.1** | **100.0** |
| Trojan | 74.3 | 100.0 | 78.8 | 100.0 | **91.9** | **100.0** | 71.6 | 99.8 | 71.1 | 99.5 | **87.1** | **100.0** | 77.5 | 100.0 | 79.6 | 100.0 | **93.4** | **100.0** |
| Clean label | 72.4 | 100.0 | 79.8 | 100.0 | **92.1** | **100.0** | 69.8 | 100.0 | 81.5 | 100.0 | **89.4** | **100.0** | 78.4 | 100.0 | 84.2 | 100.0 | **92.9** | **100.0** |

integrity verification are shown in Table 1. We can see that some tampering types are harder to detect than others. *The detection rate is significantly lower when model tampering is subtler.* On both CIFAR-10 and GTSRB, SSF and PulicCheck both have a significantly lower tamper detection rate for unlearning, online learning, fine-tuning, targeted attacks, and single-sample poisoning degradation attacks than that for model compression (pruning and quantization), backdoor attacks, Trojan attacks, etc. For example, SSF fails in detecting fine-tuning with a learning rate of $10^{-5}$ on these datasets. We conclude that existing SOTA fingerprinting methods are insufficiently sensitive in detecting harder-to-observe subtle tampering, calling for fingerprint samples with higher sensitivity.

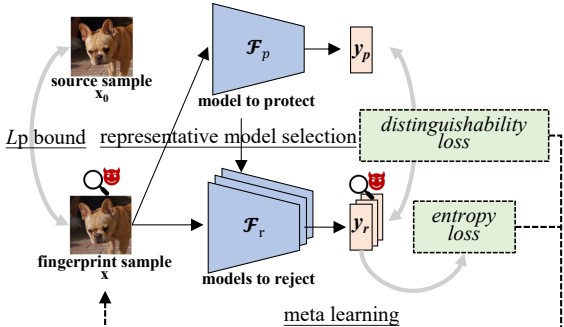

**Figure 1: Overview of MiSentry's fingerprint generation.**

## 4 MISENTRY: FINGERPRINTING DNNS WITH HIGHER SENSITIVITY

In this section, we first describe the threat model and present desirable properties of fingerprint samples. To achieve desirable properties, we formalize the generation of fingerprint samples as an optimization problem to distinguish two models, select representative models to construct a model zoo to maximize detection sensitivity and leverage meta-learning to make fingerprint samples generated on a small number of models effective on unseen models. MiSentry's fingerprint generation is outlined in Fig. 1.

### 4.1 Threat Model

Like SSF [22], we assume white-box fingerprint generation and black-box integrity verification. Specifically, a model is under white-box access when its fingerprint samples are generated. When a model's integrity is verified, we assume the model cannot be accessed directly. Integrity verification can be conducted only by querying the model with samples. When queried, a DNN model returns only the top-1 label, without returning its confidence level or the prediction vector.

MiSentry can be utilized for both private and public integrity verification. The detection performance remains consistent between public and private integrity verification. The primary difference between the two types of integrity verification lies in the fingerprint samples' usage: In public verification as shown in Fig. 2, model owners generate fingerprint samples and send them to a trusted

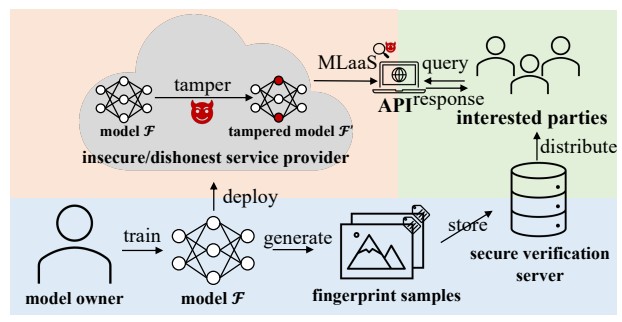

**Figure 2: Public integrity verification [58].**

third party, from which interested parties can request samples to verify a cloud model's integrity.

In contrast, in a private setting, only the model owner or authorized entities can use the generated fingerprint samples provided by the model owner for verification. It is important to note that public verification makes integrity verification more practical but significantly increases the risk of fingerprint sample leakage due to the involvement of untrusted users.

In addition, we assume that the DNN service provider is not trustworthy: it may detect probing fingerprint samples and evade tampering detection. We also assume that adversaries can get access to fingerprint samples already used for verifying integrity and exploit them to evade tampering detection.

### 4.2 Desirable Properties

With the above threat model, fingerprinting samples should have the following desirable properties:

- **Sensitivity.** Fingerprint samples should be sensitive to any practically observable modifications (see Section 3.1 for details) to the target model.
- **Uniqueness.** Fingerprint samples should be able to distinguish the target model from other models of the same or different architectures, trained with the same or different datasets, that perform the same task as the target model. This property ensures the uniqueness of the target model, i.e., it cannot be replaced by a similarly trained model. It is unexplored by existing methods.
- **Covertness.** To ensure that fingerprint queries can't be distinguished from normal queries and prevent malicious cloud providers from identifying fingerprint queries to evade integrity verification, a fingerprint sample should not be abnormal. As proposed by SSF [22], a fingerprint sample should look natural. We call this property *input covertness*. We extend the covertness of input images to output prediction vectors: the prediction vector of a fingerprint sample should also be similar to that of normal samples, denoted as *output covertness*. Both SSF [22] and PublicCheck [58] require the input covertness but do not require output covertness.
- **Robustness to Partial Leakage.** Fingerprint samples already used in verifying integrity may be accessible to adversaries, who can use them to adjust the model to evade tampering detection. The remaining fingerprint samples should be effective in detecting such adaptively modified models.
- **Generalizability.** A fingerprinting method should be generalizable to different models and datasets. In addition, fingerprint

samples should be effective in detecting all potential practical modifications to DNN models. For MiSentry, the latter means that fingerprint samples should be effective in distinguishing the target model from not only the tampered models in the model zoo but also tampered models with unseen tampering types.

### 4.3 Formalizing as an Optimization Problem

**Natural looking.** Covertness requires a fingerprint sample to look natural. To fulfill this requirement, we generate a fingerprint sample $x$ from a normal sample $x_0$ with a distortion bound: $||x - x_0||_p \le \epsilon$ for some norm $p$.

**Distinguishablity loss.** Suppose we have two models, $\mathcal{F}_p$ and $\mathcal{F}_r$, where $\mathcal{F}_p$ is the target model (i.e., to protect) and $\mathcal{F}_r$ is a model to reject (i.e., to distinguish from the target model). To generate a fingerprint sample $x$ to distinguish them, we maximize the divergence of the output predictions of the two models over input $x$: $Divergence(\mathcal{F}_p(x), \mathcal{F}_r(x))$, which can be measured by Kullback-Leibler (KL) divergence. The distinguishability loss corresponding to this inter-model divergence is thus:

$$\ell_D(x, \mathcal{F}_p, \mathcal{F}_r) = -D_{KL}(\mathcal{F}_p(x)||\mathcal{F}_r(x))$$
$$= \sum_j \mathcal{F}_p(x)_j \ln \frac{\mathcal{F}_r(x)_j}{\mathcal{F}_p(x)_j} \qquad (1)$$

where $\mathcal{F}_p(x)_j$ is the $j$-th element in the prediction vector of model $\mathcal{F}_p$ on input $x$. Note that we have taken a negative sign before the KL divergence in Eq. 1 since we minimize this loss term.

**Entropy loss.** The entropy loss is designed to ensure output covertness, which makes MiSentry the only one among the three methods considered that meets the output covertness criterion, as demonstrated in Fig. 5 in Section 5.3. Output covertness ensures that the prediction vectors of fingerprint samples are similar to those of normal samples, making fingerprinting queries indistinguishable from normal queries based on model outputs. This prevents a malicious cloud service provider from identifying fingerprinting queries to evade integrity verification. On $\mathcal{F}_p$, a normal sample has likely a high probability on its true label and low probabilities on others. We also expect that $\mathcal{F}_r$ behaves similarly to $\mathcal{F}_p$, otherwise it can be easily differentiated from $\mathcal{F}_p$ without relying on a fingerprinting method. Covertness requires that the prediction vector of a fingerprint sample $x$ is indistinguishable from that of a normal sample on both $\mathcal{F}_p$ and $\mathcal{F}_r$. To fulfill this requirement, we minimize the information entropy H of the prediction of $x$ on both $\mathcal{F}_p$ and $\mathcal{F}_r$:

$$\ell_{E_\gamma}(x, \mathcal{F}_\gamma) = H(\mathcal{F}_\gamma(x)) = -\sum_j \mathcal{F}_\gamma(x)_j \log \mathcal{F}_\gamma(x)_j \qquad (2)$$

where $\gamma \in \{p, r\}$ for models $\mathcal{F}_p$ and $\mathcal{F}_r$, respectively. In addition, we calculate the entropy loss of the normal samples with the same label as $x$ on each model and determine a distribution range $[\alpha_\gamma, \beta_\gamma]$, e.g., $10^{th}$ percentile to $90^{th}$ percentile. We require that the entropy loss of $x$ is within the range on both models: $\alpha_\gamma \le \ell_{E_\gamma} \le \beta_\gamma, \gamma \in \{p, r\}$.

**Optimization problem.** A fingerprint sample $x$ is generated by meeting the above requirements simultaneously, i.e., by solving the following optimization problem:

$$\min_x \ell_{total}(x, \mathcal{F}_p, \mathcal{F}_r)$$
$$\text{s.t. } ||x - x_0||_p \le \epsilon, \alpha_p \le \ell_{E_p} \le \beta_p, \alpha_r \le \ell_{E_r} \le \beta_r \qquad (3)$$

with

$$\ell_{total}(x, \mathcal{F}_p, \mathcal{F}_r) = \ell_D(x, \mathcal{F}_p, \mathcal{F}_r) + \lambda_1 \cdot \ell_{E_p}(x, \mathcal{F}_p) \\ + \lambda_2 \cdot \ell_{E_r}(x, \mathcal{F}_r) \tag{4}$$

where $\lambda_1$ and $\lambda_2$ are two weighting hyperparameters of different loss terms. Eq. 3 can be solved iteratively with the projected gradient descent like in PGD [32, 33]:

$$x_{k+1} = Clip_\epsilon(x_k - \eta \cdot \nabla_{x_k} \ell_{total}(x_k, \mathcal{F}_p, \mathcal{F}_r)) \tag{5}$$

**Generation diversity.** We generate our fingerprint samples starting from different source samples. It can generate diverse fingerprint samples widely spread all over the decision boundary of the original model. Each fingerprint sample is located in a distinct local region, which is independent and unpredictable. Thus leaked fingerprint samples can hardly influence the effectiveness of unleaked ones.

## 4.4 Representative Model Selection

We next describe the representative model selection in MiSentry to maximize the detection sensitivity of the generated fingerprint samples while minimizing the number of models selected for fingerprint generation. To make MiSentry more practical, we require the models in the model zoo of meta-learning to be of the same architecture and trained with the same dataset as the target model.

In support vector machines (SVMs) [56], support vectors are the instances that result in the maximal margin and are closest and crucial to the decision boundary of the hyperplane. A model could be tampered with in countless ways, but limited (and preferably as few as possible) tampering instances can be used to generate fingerprint samples. Similar to the support vectors, crucial tampering instances should be identified to generate fingerprint samples capable of distinguishing all practically observable modifications.

Intuitively, *fingerprint samples generated using subtler modifications should likely be capable of distinguishing more significant modifications and exhibit higher detection sensitivity.* The rationale behind this intuition is that a model modified more subtly has a decision boundary closer to that of the target model. Among models subjected to the same type of modification, the subtlest one typically exhibits the smallest deviation, such as the least samples' prediction variation or the fewest required training epochs, since tampering typically demands a certain number of training samples and steps.

To validate the intuition, we conduct the following study: we generate fingerprint samples using models fine-tuned over varying numbers of epochs (1 and 10) and use these samples to detect modifications across a range from 1 to 15 epochs. Additionally, we generate fingerprint samples using models updated via online learning with 1 and 10 updated samples and used these to detect updates in models with up to 10 samples. Our findings from experiments on the CIFAR10 dataset, illustrated in Fig. 3, demonstrate that fingerprint samples from the subtlest modifications are more effective in detecting larger modifications and possess the highest detection sensitivity.

These results suggest that subtler model modifications are more crucial for the generation of effective fingerprint samples, being closer to the detection hyperplane. Consequently, we incorporate

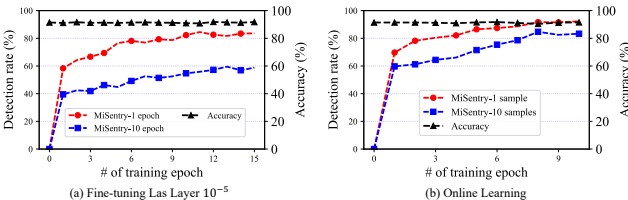

(a) Fine-tuning Las Layer $10^{-5}$     (b) Online Learning

**Figure 3: Tamper detection rate w.r.t. the number of training epoch/samples on CIFAR10.**

these subtlest model modifications into our model zoo as "support vectors" for generating fingerprint samples.

For models with different types of modifications, empirical judgment is used to determine which type likely produces fewer perturbations to the decision boundary. It is generally harder to distinguish models of the same architecture and trained with the same dataset than models of different architectures or trained with different datasets. That intuition leads to selecting small variations of models of the same architecture and training with the same dataset as the target model into the model zoo for meta-learning (see Appendix C.3 for details).

This representative model selection of MiSentry makes fingerprint sample generation more practical since it does not require any model trained with extra data, which may be inaccessible to the model owner due to privacy protection.

In addition, by leveraging meta-learning to enhance generalizability [62](see Appendix A for more details), this limited selection of models does not compromise the effectiveness of generated fingerprint samples in detecting unseen tampering types. As shown in Tab. 1, fingerprint samples generated by MiSentry are highly generalizable: they are effective in distinguishing from the target model not only the models in the model zoo but also models unseen in their generation.

## 5 EXPERIMENTAL EVALUATION

In this section, we evaluate our MiSentry's performance using BIVBench following the same settings as Section 3.2 and conduct additional evaluations on leakage robustness, output covertness, detection sensitivity, each component's effect, and time cost.

## 5.1 Tampering Detection Performance

MiSentry's detection performance is also presented in Tab. 1. We can make the following observations.

**MiSentry's fingerprint samples are generalizable**. Even though MiSentry uses a few types of model modifications in the model zoo of meta-learning to generate fingerprint samples, generated fingerprint samples are effective in distinguishing unseen tampering types and independently trained models from the target model. The tampering detection rates of MiSentry are all above 36% using a single fingerprint sample and above 90% using 5 fingerprint samples for the three datasets.

**MiSentry outperforms existing methods**. MiSentry has a reasonable tampering detection rate even for the hardest-to-detect tampering types. It outperforms both SSF and PulicCheck in general and by a large margin for subtler tampering types.

**Detection rate of a larger-scale dataset and model is generally higher**. The detection rates of subtler tampering types,

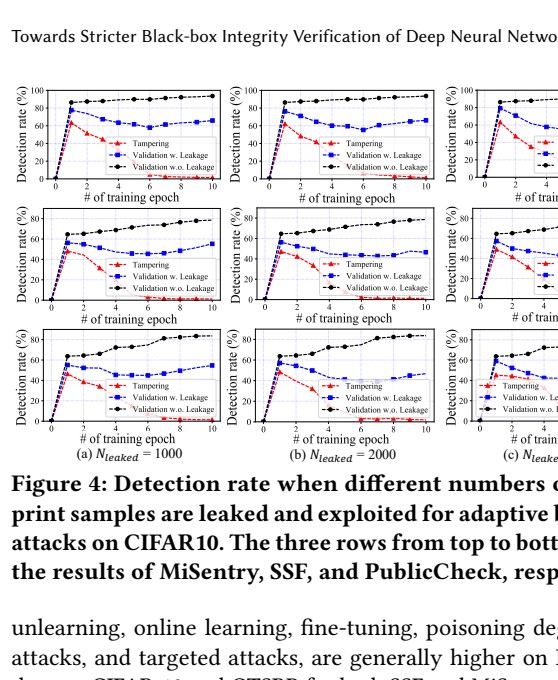

**Figure 4: Detection rate when different numbers of fingerprint samples are leaked and exploited for adaptive backdoor attacks on CIFAR10. The three rows from top to bottom show the results of MiSentry, SSF, and PublicCheck, respectively.**

unlearning, online learning, fine-tuning, poisoning degradation attacks, and targeted attacks, are generally higher on ImageNet than on CIFAR-10 and GTSRB for both SSF and MiSentry. It can be explained as follows. For fine-tuning, a model is fine-tuned with 1 epoch. Since ImageNet has many more training samples than the other two datasets, fine-tuning an ImageNet model incurs more severe modifications than fine-tuning CIFAR-10 and GTSRB models, making it easier to be detected. For other tampering types, the modification adjusts more model parameters on an ImageNet model than on a model of CIFAR10 or GTSRB since the ImageNet model is much larger, which makes tampering detection easier.

SSF-BO [34], a variant of SSF, uses Bayesian optimization to generate fingerprint samples. SSF-BO is infeasible for large-scale datasets like ImageNet. We use the code of SSF-BO from [42] to detect model tampering on MNIST [35], a handwritten digits dataset. The results can be found in Appendix. F. MiSentry surpasses SSF-BO in detecting all the tested model tampering.

## 5.2 Adaptive Attacks with Leaked Samples

In public integrity verification, an adversary can collect already used fingerprint samples and use them for an adversarial tampering attack by pursuing a model manipulation goal while preserving the top-1 labels on collected fingerprint samples simultaneously. To evaluate existing SOTA fingerprint methods and MiSentry's robustness to such adaptive attacks, we assume that $N_{leaked}$ already used fingerprint samples are collected and exploited by the attacker, where $N_{leaked}$ represents the number of leaked fingerprint samples, and evaluate the detection performance of the three methods on newly-generated unused fingerprint samples. The leaked fingerprint samples comprise *tampering set* which is used in the adversarial tampering attacks to evade MiSentry detection, while the unleaked fingerprint samples comprise *validation set* which is used for integrity verification.

To investigate the effectiveness of existing fingerprint methods and MiSentry in such increasingly threatening scenarios, we increase the number of leaked fingerprinting samples, $N_{leaked}$, from 1,000 to 3,000. Here, we conduct adaptive tampering attacks with backdoor attacks and last-layer fine-tuning respectively since they are the most common malicious and benign model modifications.

Experimental results of adaptive backdoor attacks are shown in Fig. 4 and those of adaptive last-layer fine-tuning are presented in Appendix E. The black dot line shows the detection rate when no leaked fingerprint samples are exploited, while the blue square line shows the detection rate when $N_{leaked}$ leaked fingerprint samples are exploited for an adaptive tampering attack. The red triangle line shows the detection rate of the leaked fingerprint samples (i.e., on the tampering set). We can conclude that:

• For all three methods, more leaked fingerprinting samples could lead to a reduced detection rate. For the adaptive backdoor attacks, when $N_{leaked}$ is 1000, the detection rates of MiSentry, SSF, and PublicCheck can drop to as low as 58.9%, 45.4%, and 44.9%, respectively. When $N_{leaked}$ increases to 3000, the lowest detection rates of MiSentry, SSF, and PublicCheck can further decline to 52.8%, 38.9%, and 39.6%, respectively. A similar tendency can be observed for the adaptive last-layer fine-tuning. However, further increases in the number of leaked fingerprinting samples can no longer significantly degrade the detection performance.

• The detection rate of MiSenrty consistently remains higher than that of SSF and PublicCheck in all cases. When $N_{leaked}$ is 1000, the lowest detection rate of MiSentry is 1.28× that of SSF and 1.31× that of PublicCheck under such an adaptive backdoor attack, respectively. A similar advantage still persists when $N_{leaked}$ is 3000. For the adaptive fine-tuning attack, we find this advantage of the detection rate of MiSenrty to that of PublickCheck enlarges to 2.25×.

• While more leaked fingerprinting samples could lead to a reduced detection rate, the detection performance of our MiSentry remained reasonably robust. The detection rate of MiSentry remains consistently higher than 50% and 29% under the backdoor and fine-tuning attack respectively. It is worth noting that SSF is almost incapable of detecting whether the model has been tampered with in the early stages of the adaptive fine-tuning attack. Only with the increase of fine-tuning training epochs does SSF gradually gain the ability to detect potential tampering, albeit the detection rate remains very low, indicating inherently low sensitivity of SSF.

The robustness of detection performance to leakage can be attributed to generation diversity. Starting from different source samples, our MiSentry can generate diverse fingerprint samples widely spread all over the decision boundary of the original model. Each generated fingerprint sample is independent and unpredictable since it is located in a distinct local region. Thus leaked fingerprint samples can hardly influence the effectiveness of unleaked ones.

## 5.3 Perceptual Quality and Output Covertness

Appendix D shows some fingerprint samples generated by MiSentry, SSF, and PublicCheck on the three datasets. These fingerprint samples exhibit a natural appearance, and the perturbations they introduce to the source samples are almost indiscernible. MiSentry and SSF show similar perceptual quality to the original samples, with noise-like artifacts, while PublicCheck's artifacts involve altered texture and color temperature. Their perceptual quality is comparable overall.

To avoid being singled out, the output of a fingerprint sample should be indistinguishable from that of a normal sample. Fig. 5 shows the distribution of prediction probabilities of top-1 labels for

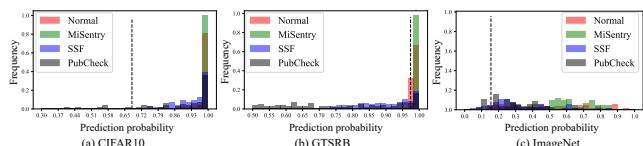

Figure 5: Distributions of prediction probabilities of top-1 labels for normal samples and fingerprint samples.

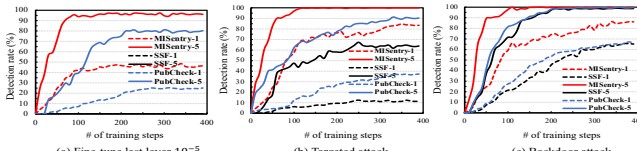

Figure 6: Detection rates vs the number of training steps on CIFAR10.

fingerprint samples generated by MiSentry, SSF, and PublicCheck and normal samples in the test set. We can see that the prediction probabilities of MiSentry are within those of normal samples, while some prediction probabilities of SSF and PublicCheck fall into a region below the bottom 5% low confidence of normal samples (shown by the dashed lines), especially for GTSRB, which makes integrity probing using fingerprint samples generated by SSF and Public-Check potentially singled out. Only MiSentry meets the output covertness requirement among the three fingerprinting methods.

## 5.4 Detection Sensitivity

During the manipulation of the target model, we would like to know how quickly a fingerprinting method can detect the manipulation. We explore this detection sensitivity, i.e., the relationship between tampering detection rate and magnitude of modification starting from the initial untampered model. Fig. 6 shows the tampering detection rates of different fingerprint methods at different training steps on CIFAR10 for fine-tuning the last layer with a learning rate of $10^{-5}$, targeted attacks, and backdoor attacks. We can make the following observations. First, the detection rate of MiSentry increases more rapidly than that of SSF and PublicCheck, indicating that MiSentry has higher detection sensitivity than SSF. Second, when using 5 fingerprint samples for tampering detection, the detection performance of MiSentry saturates around 100 training steps. Third, SSF fails to detect the fine-tuning, which agrees with that reported in Section 3.2.

## 5.5 Ablation Study

We conduct an ablation study for each component of our MiSentry to analyze their respective effects on CIFAR10. Tab. 2 presents the tampering detection rates using a single fingerprint sample for different model zoo constructions. Additional results are shown in Appendix G. We can make the following important observations.

• Subtly modified models (i.e., last-layer fine-tuning) are crucial for detecting all modification types, while other modification types in the model zoo mainly enhance detection performance for the same type.

• The generalization ability of fingerprint samples improves with an increasing number of representative models per tampering type but saturates when the number is larger than 6.

Table 2: Ablation study on modification types for constructing model zoo. FT stands for fine-tuning.

| Tampering\Model zoo | $FT + Retrain$ | $FT$ | $FT_{Last}$ | $FT_{All}$ |
|---|---|---|---|---|
| Degradation-C | 91.3 | 91.1 | 88.9 | 79.4 |
| Targeted Attack | 83.8 | 84.2 | 79.3 | 77.2 |
| Backdoor | 86.4 | 85.9 | 81.5 | 78.1 |
| Quantization | 90.7 | 91.2 | 89.4 | 80.1 |
| Pruning | 89.5 | 87.4 | 78.3 | 72.4 |
| Fine-tuning$_{Last}$ | 73.7 | 73.8 | 75.8 | 48.9 |
| Fine-tuning$_{All}$ | 59.2 | 59.4 | 49.2 | 48.1 |

Table 3: Average time cost per model construction or fingerprint sample generation (in seconds).

| | | CIFAR10 | GTSRB | ImageNet |
|---|---|---|---|---|
| Tampered model | Fine-tuning$_{Last}$ | 9.5 | 10.4 | 4980.3 |
| | Fine-tuning$_{All}$ | 11.2 | 12.8 | 6361.5 |
| | Retraining$_{All}$ | 1473.6 | 1519.8 | N/A |
| Fingerprint sample generation | | 95.7 | 86.1 | 925.2 |

• Without the $L_p$ norm bound, the generated fingerprints are noise-like images, which can be easily distinguished from normal samples. However, these noise-like fingerprints have higher detection sensitivity (tampering detection increases up to 0.359×).

• Without entropy loss, the generated fingerprint samples are more likely to have abnormal prediction vectors, with AUC scores decreasing from 0.651 to 0.397 when detecting fingerprint samples by their confidence scores.

• MiSentry exhibits similar defection performance across different model architectures of the target model, confirming its generalizability.

## 5.6 Time Cost

We use a single Nvidia RTX3090 GPU in our experiments. The average time cost of constructing a model for each tampering type and fingerprint sample generation of our MiSentry is shown in Tab. 3. Overall, the time cost of MiSentry increases with the increase in model parameter size. These time costs are acceptable compared to the benefits brought by integrity verification.

## 6 CONCLUSION

In this paper, we have introduced significant advancements in the field of black-box integrity verification (BIV) for DNN models. We pioneer a stricter form of BIV by focusing on subtle tampering and defining observable modifications, setting new standards for BIV. We develop BIVBench, the first benchmark tailored for stricter BIV. It includes 16 types of tampering scenarios, significantly more than previous studies, revealing the limitations of existing fingerprinting methods. We also introduce MiSentry, a novel fingerprinting method using a refined set of subtly tampered models to produce highly sensitive and effective fingerprint samples. MiSentry has proven superior in detecting tampering, even those unseen during its generation. To conclude, we establish new benchmarks and methodology for the BIV of DNN models, paving the way for further studies in this crucial field.

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
