# OpenReview forum: "Towards Stricter Black-box Integrity Verification of Deep Neural Network Models"
_acmmm.org/ACMMM/2024/Conference — MM2024 Poster_

### Official Review · Reviewer_Ksdt · 2024-04-27

**Rating:** 2
**Confidence:** 4

**Summary:**

This paper studies black-box integrity verification of DNN models.
This paper focuses on "subtle modifications" as a series of integrity violations that are harder to be identified than ordinary integrity violations (such as backdoor inserting) and includes many subtle modfications into the threat model, named after BIVBench.
Empirical results on BIVBench show that some SOTA intergrity verification schemes turn out to be uncapable of detecting subtle modifications (in Table 1).
This paper further proposes a new black-box DNN integrity verification scheme, which is another fragile fingerprinting scheme that is non-invasive and easy to be applied.
The triggers are found by leveraging surrogate DNN models (the authors used the term "model zoo" for this concept) which have undergone subtle modifications.
Concretely, triggers are optimized so that their outputs are differentiated in the original model and surrogate models.
Experiments demonstrated that triggers found by this scheme(MiSentry) turn out to be more sensitive, general, and covert.

**Strengths:**

1. This paper explicitly mentioned and discussed the topic of "subtle modifications" as a challenge to current fragile fingerprint schemes for DNN models and verified that existing schemes might be unreliable. I believe that this point is very important and inspiring.

2. The presentation is very well, I felt comfortable and smooth in following the authors' intuition and reasoning.

3. I think that the authors have covered a broad range of metrics that are used to evaluate fragile fingerprint for DNN models (sensitivity, uniqueness, covertness, robustness, generalizability, and ablation studies), I appreciate these comprehensive discussions that prevented conclusions from multiple perspectives.

**Limitations:**

I have several concerns about the proposed scheme's (MiSentry) technical soundness and fairness. Some parts of the discussions contained in this paper are unnecessary or confusing.

1. On the technical novelty, it seems to me that MiSentry tries to find some partially adversarial samples (I mean normal samples added with a slight perturbation, which moves the samples towards the decision boundary but not over it) so that their outputs are different in the original DNN model and surrogate models, which are modified version of the original model. (I might be wrong since Sec.4.4-paragraph 1 said "be of the same architecture and trained with the same dataset", and I believe that using modified versions of the original model results in more subtle modifications) What makes it essentially different from an adversarial attack, which has already been widely adopted to find samples near the decision boundary and are thus sensitive? I believe that PublicCheck used the same intuition, although it adopted a generator rather than gradient-based adversarial sample attack, so what is the theoretical or conceptual improvement?

2. On the contribution's significance. I believe that modifications are nonetheless going to be more subtle in the future, with the emergence of LoRA for LLMs and text inversion for Diffusion Models. The contribution of proposing a collection of modifications such as BIVBench is questionable, even if it is the first of their kind as claimed by authors. There has to be more formal and mathematical discussions on the level of subtleness and the performance limitations of MiSentry (why it is hard or impossible to design stricter schemes?).

3. On the scheme's soundness. I found that you used Fine-Tuning_{Last}-5 as surrogate models in Appendix C.3 (I believe that this point has to be incorporated into the main text since this is the most important setting). Now Fine-Tuning_{Last}-5 is already the strongest (or the most "subtle") modification according to Table 1, so samples that are trained to differentiate models undergone Fine-Tuning_{Last}-5 and the original model can for sure detect other weaker attacks. This comparison is unfair. Other baselines did not explicitly incorporate as much surrogate models as MiSentry. Can the generalizability be reduced to training the triggers on a hard task and test them on an easy task? Appendix G.1 Table 5 seems to support this statement, since surrogate model/model zoo heavily affect the detection rate. The comparitive advantages seem to be a cherry-picking result: you can always include models undergone more subtle modifications (e.g., fine-tuning less parameters with a smaller learning rate) to run MiSentry, and report improved statistics on less subtle modifications.

4. In Sec.4.2, "Uniqueness" is not a desirable property for fragile fingerprints, it is desirable only for robust fingerprints (robust fingerprints are for ownership verification where sensitivity is the least favorable property). This is why exisitng literatures did not explore this aspect. After all, the authors provided no further examination of their scheme regarding uniqueness in the rest of their paper.

5. In Sec.4.2, "Robustness to Partial Leakage". This is not a concern that worths discussions,  you can always produce new triggers in the context of a fingerprinting scheme. Using a collection of new triggers during each round of integrity verification is enough. The leakage is a problem only for watermarking schems (where a finite list of triggers have to be determined beforehand), or when only a finite collections of triggers can be produced, or when the cost of randomly producing new triggers is prohibitive (Sec.5.6 suggest that this is cheap).

6. Is there any results on the performance of triggers regarding preprocessing? (e.g., the adversary applies JPEG-compression to triggers)

**Suitability:**

2

---

### Official Review · Reviewer_21Fd · 2024-05-16

**Rating:** 4
**Confidence:** 4

**Summary:**

The main work of this paper consists of two aspects:
firstly, the design of a new fragile fingerprint called MiSentry, which is used to verify the integrity of models;
secondly, the creation of the first benchmark for verifying model integrity in black-box scenarios, named BIVBench.

The paper validates the fragility of MiSentry through BIVBench and compares it with previous state-of-the-art (SOTA) fragile fingerprints,
demonstrating the superiority of the proposed method.

**Strengths:**

The proposed MiSentry is highly sensitive to various changes in the model, especially those that result from modifying even a small portion of the model parameters, which leaves a deep impression on me.

The design of BIVBench is very comprehensive, taking into account almost all possible modifications to the model that could occur in real-world scenarios.

The experiments are thorough, testing the fingerprints trained on multiple datasets and models.

**Limitations:**

1. It seems that the effectiveness of the fingerprint detection is significantly reduced in the case of adaptive attacks, where the fingerprint is leaked. Would this fragile fingerprint method also be less effective on models that have undergone adversarial training? I hope to see related experiments on this matter.

2. Generating the fingerprint requires training within a model zoo. In this context, would the cost of generating the fingerprint be too high?

3. The BIVBench proposed in this paper distinguishes between benign and malicious modifications. Is it necessary to make this distinction?

4. Have there been tests on models other than classification models, or models other than convolutional models? For example, Vision Transformers (ViT).

**Suitability:**

3

---

### Official Review · Reviewer_6KCx · 2024-05-17

**Rating:** 5
**Confidence:** 3

**Summary:**

The paper introduces BIVBench, a benchmark for black-box integrity verification of DNN models, and MiSentry, a novel fingerprinting method. BIVBench encompasses 16 observable model modifications, highlighting the inadequacy of existing methods in detecting subtle tampering. MiSentry maximizes prediction divergence between target and tampered models to generate sensitive fingerprint samples. It outperforms existing methods, especially in detecting subtle tampering, enhancing DNN model security in cloud services.

**Strengths:**

1. This paper is the first to explore black-box integrity verification of DNN models in a stricter form by defining observable modification and considering subtle tampering. Introduces MiSentry, a novel fingerprinting method, and BIVBench, the first benchmark specifically designed for black-box integrity verification of DNN models.
2. The motivation is clear. Stricter integrity verification for detecting harder-to-observe modifications is useful, since an end user may need to check if the model he/she is using is really the model he/she wants to use.
3. The paper is well-written, providing clear explanations of the problem, good motivating experiments, and intuitive solutions. The paper is overall easy to follow.
4. The evaluation is comprehensive and the results are effective. MiSentry outperforms existing state-of-the-art fingerprinting methods, particularly in detecting subtle tampering.

**Limitations:**

Overall I think this work is good and suitable to be published at ACM MM. I do not find obvious mistakes or weaknesses.

**Suitability:**

2

---

### Official Review · Reviewer_rKGd · 2024-05-27

**Rating:** 4
**Confidence:** 2

**Summary:**

This paper introduces BIVBench, the first benchmark for black-box integrity verification of deep neural networks, covering 16 tampering scenarios.

**Strengths:**

It highlights the limitations of current methods in detecting subtle tampering and proposes MiSentry, a new approach using meta-learning and a model zoo to generate highly sensitive fingerprint samples. Extensive evaluations show MiSentry significantly outperforms existing methods in detecting both subtle and significant tampering.

**Limitations:**

1. While MiSentry's use of meta-learning is innovative, the paper could benefit from a deeper exploration of the limitations and challenges associated with this approach. For instance, the computational overhead and the potential difficulties in selecting an appropriate model zoo could be further discussed.

2. Although the paper provides extensive empirical evaluations, additional real-world validation would strengthen the claims. Implementing and testing MiSentry in live cloud environments would provide valuable insights into its practical effectiveness and potential challenges.

3. The comparison with existing methods is comprehensive, but the paper could include a broader range of baselines, particularly those from related fields such as adversarial robustness and secure model deployment. This would provide a more holistic view of MiSentry's relative strengths and weaknesses

4. The sensitivity of MiSentry to various hyperparameters, such as the selection of models in the model zoo and the parameters for meta-learning, could be elaborated upon. Understanding how these factors influence performance would aid in the practical adoption and fine-tuning of the method.

**Suitability:**

2

---

### Meta-Review · Area_Chair_KKEs · 2024-06-28

**Recommendation:** Accept (Poster)
**Confidence:** 4

**Metareview:**

The paper introduces a benchmark for black-box integrity verification of DNN models, highlighting the inadequacy of existing methods in detecting 'subtle' modifications. Besides, the authors propose a new fragile fingerprinting scheme that is non-invasive and highly effective. The reviewers generally recognize the problem's significance, the extensive benchmark, and the paper writing. However, the reviewers still have some critical concerns, especially those regarding the technical contributions and model zoo selection. The four reviewers engaged in an intense discussion during the post-rebuttal phase but they failed to ultimately agreed upon. This brings the score of this paper to the borderline. Note that Reviewer Ksdt has corrected the misunderstanding and provided further explanations at this moment.

After receiving the rebuttal message from the authors, I contacted Reviewer Ksdt to re-evaluate the previous comments. I have also carefully read all materials, including the submission, all comments, the author's rebuttal, and the reviewer's discussions. Although this paper still has some shortcomings (especially the details and settings for model zoo selection), I think this paper has made sufficient contributions to this field. By trading off the strengths and weaknesses of this paper, I decide to accept this paper. However, I hope the authors can carefully read the post-rebuttal comments from Reviewer 6KCx and try to address them in the final version of this paper. I think it can greatly improve this paper.